# Electronic Cigarettes and Youth in the United States: A Call to Action (at the Local, National and Global Levels)

**DOI:** 10.3390/children6020030

**Published:** 2019-02-20

**Authors:** Brian P. Jenssen, Rachel Boykan

**Affiliations:** 1Department of Pediatrics, University of Pennsylvania School of Medicine and PolicyLab and the Center for Pediatric Clinical Effectiveness, Children’s Hospital of Philadelphia, Philadelphia, PA 19104, USA; 2Department of Pediatrics, Renaissance School of Medicine at Stony Brook University, Stony Brook, NY 11794, USA; rachel.boykan@stonybrookmedicine.edu

**Keywords:** e-cigarette, vape devices, tobacco, nicotine, secondhand smoke exposure

## Abstract

E-cigarettes have emerged and soared in popularity in the past ten years, making them the most common tobacco product used among youth in the United States (US). In this review, we discuss what the Surgeon General has called a public health “epidemic”—the precipitous increase in youth use of e-cigarettes and the health consequences of this behavior. Further, we review tobacco control policy efforts (e.g., Tobacco 21, banning flavors, advertising restrictions, and clean indoor air laws)—efforts proven to be critical in reducing cigarette smoking and smoking-related disease and death among US children and adults—including their potential and challenges regarding managing and mitigating the emergence of e-cigarettes. Finally, we close with a discussion of the efforts of transnational tobacco companies to rebrand themselves using e-cigarettes and other new products.

## 1. A Brief History of E-Cigarettes

E-cigarettes have emerged and soared in popularity in the past ten years, making them the most common tobacco product used among youth. The invention (in 2003), and patent of the first e-cigarette (in 2007) is generally attributed to a Chinese pharmacist, Hon Lik [1]. However, designs for a device to aerosolize nicotine may be traced back to the early 1960s, with the development of British American Tobacco’s Project Ariel [2]. As early as 1959, tobacco company research recognized (though not publicly acknowledged) nicotine’s addictive nature and, therefore, the importance of nicotine in maintaining a customer base [1]. As emerging research linked smoking with cancer, it seemed worthwhile to invest in the development of a potentially less harmful nicotine delivery system. Ariel was designed as a cylinder within an outer layer: an outer mixture of tobacco would heat up the inner part and nicotine, either coated on the walls of the tube or in liquid form, would be vaporized [2]. However, despite two patents, Project Ariel never made it to the market; this was perhaps because the design was imperfect, and perhaps because—despite the science linking smoking with cancer—cigarette smoking was still immensely popular and unregulated. A newer, safer, cigarette may not have seemed as necessary after all, and may have been perceived as competing with traditional cigarettes, the established product. Twenty-five years later, Philip Morris also explored similar technology, with the development of their “Premier” (1988) Capillary Aerosol Generator (CAG), in which a pump forced liquid into a small capillary tube, which was then heated and aerosolized. Though this product also did not make it to market, the technology is very similar to Philip Morris’ subsequent 2009 electronic cigarette patent [3].

## 2. E-Cigarettes: An Epidemic and a Threat to Public Health

Since their emergence on the market in 2007, e-cigarettes have evolved from the initial first generation (“cig-a-like”) products to sophisticated and greatly varied, albeit still largely unregulated, devices. At the time of this writing, there is evidence that e-cigarettes may be safer in some ways than combusted cigarettes (no combustion, no tar) though their potential as a true “harm-reduction” product has yet to be determined [4]. Whether or not e-cigarettes can help adult smokers quit smoking is controversial and beyond the scope of this paper. We will discuss what the Surgeon General in the United States has called a public health “epidemic”—the precipitous increase in youth use of e-cigarettes and the health consequences of this behavior [5].

Before e-cigarettes gained popularity, smoking among high-school students had started to decline significantly, from 28% in 1996–1997 to 15.8% in 2011 to 7.6% in 2017 [6,7]. This decline—a true public health triumph—may be attributed to multiple tobacco control measures, including restrictions on tobacco company advertising, promotions and sponsorship, taxation of tobacco products, clean indoor air laws, banning of flavorings in cigarettes (except menthol), and decreased access of cigarettes to teenagers through the adoption of local and statewide Tobacco 21 legislation (raising the legal age of sale from 18 to 21) [8]. Indeed, in the past decades, attitudes have changed: teenagers recognize the dangers of smoking and no longer find it attractive or “cool.” By contrast, e-cigarettes, now used by over 20% of high-school students, are not viewed by adolescents as dangerous or even comparable to cigarettes [9,10,11,12,13]. USB-like e-cigarettes, and one product in particular, Juul, has captured over 75% of the market share of the e-cigarette category since its introduction in 2014–2015. This product transformed both the category and use patterns given its efficient delivery of highly concentrated nicotine, flavors, and covert design [14,15,16]. The increasing use of e-cigarettes among youth threatens five decades of public health gains in successfully deglamorizing, restricting, and decreasing the use of tobacco products [17].

Below, we will discuss what is known to date regarding the health effects of adolescent e-cigarette use. We will present evidence-based approaches to e-cigarette regulation based on established tobacco control legislation. Finally, as e-cigarettes are a global issue, we will briefly discuss e-cigarettes in the international arena.

## 3. Health Impact of E-Cigarettes on Youth

### 3.1. Inhalation of E-Cigarette Aerosol

Inhalation of e-cigarette aerosol is not benign: recent research presents concerning data regarding the potential short- and long-term health sequelae of inhaling the many varied and as of yet, unregulated, constituents of e-liquid [18,19,20]. Propylene glycol (PG) and vegetable glycerin (VG), common e-liquid humectants, carry the FDA classification of generally recognized as safe (GRAS) for ingestion or use in cosmetics, but not for inhalation. PG causes respiratory and eye irritation when inhaled, and VG forms acrolein, a respiratory irritant [4]. Flavorings, of which there are over 7000, contain several known toxic substances, including diacetyl and acetyl propionyl, which may lead to the development of bronchiolitis obliterans, or “popcorn lung” [1,21]. Aldehydes and heavy metals may be formed in the heating process, as well as potentially carcinogenic and teratogenic compounds [1,22]. Ultrafine particles found in the aerosol may contribute to systemic inflammation [1,4,23].

### 3.2. Nicotine and Nicotine Addiction

Nicotine is well-recognized as one of the most addictive substances, as addictive as heroin and cocaine [24]. Adolescents are particularly susceptible to nicotine addiction: the majority (90%) of smokers start before the age of 18, [25] a fact that has been utilized by Tobacco companies for decades in their teen-targeted advertising, marketing and even product design [26]. Adolescents may show signs of dependence with even infrequent nicotine use; sustained nicotine exposure leads to upregulation of the receptors in the prefrontal cortex, pathways which are involved in cognitive control, and which are not fully matured until the mid-twenties [27,28]. Such disruption of neural circuit development may lead to long-term cognitive and behavioral impairment and has been associated with depression and anxiety [29,30,31].

The nicotine content in e-cigarettes varies widely by product and by use. Refill solutions may contain anywhere from 1.8% nicotine (18 mg/mL) to over 5% (59 mg/mL). Nicotine delivery may be affected by the device itself, for example, by increasing the voltage which changes the aerosol delivered, or by “dripping”—a process of inhaling liquid poured directly onto coils [32]. The latest generation of e-cigarettes, “pod products,” such as Juul, have the highest nicotine content (59 mg/mL), in protonated salt, rather than the free-base nicotine form found in earlier generations, which makes it easier for less experienced users to inhale [33]. Despite the clear presence of nicotine in e-cigarettes, adolescents often do not recognize this fact, potentially fueling misperceptions about the health risks and addictive potential of e-cigarettes [34,35,36].

Are adolescents addicted to e-cigarettes? The unprecedented increase in current (past-month) users in the past year (from 11.7% of high school students in 2017 to 20.8% in 2018) [12] would imply dependence, if not addiction, given what we know about nicotine and its effects on the adolescent brain. Two recent studies utilized validated measures to identify nicotine dependence in e-cigarette using adolescents [37,38]. Indeed, exposure to nicotine from e-cigarettes may be higher than that from conventional cigarettes. In a study of adolescent pod users, their urinary cotinine (a breakdown product used to measure nicotine exposure) levels were higher than levels seen in adolescent cigarette smokers [33].

### 3.3. Progression to Cigarette Smoking and Other Drug Use

Several, separate well-designed longitudinal studies have demonstrated that adolescents who use e-cigarettes have a significantly increased risk of progressing to conventional smoking, or to subsequent co-use of combusted tobacco with e-cigarettes [39,40,41,42,43,44,45,46]. According to a recent meta-analysis, adolescents and young adults (14–30 years of age) who use e-cigarettes (compared to with those who do not) are 3.6 times more likely to report traditional cigarettes at follow-up [47]. These teenagers do not appear to be particularly at-risk for cigarette smoking, as adolescents who used e-cigarettes first appear to have fewer social and behavioral risk factors compared to traditional cigarette users [41,42,43,45]. Further, teens who use e-cigarettes are more likely to use cannabis, not just in its traditionally combusted form, but also vaporized, as e-cigarette devices present opportunities for experimentation and customization [48,49,50].

### 3.4. Second-Hand Aerosol and Other Exposures

Known harmful toxicants and carcinogens have been found in e-cigarette emissions [4,51]. These include polycyclic aromatic hydrocarbons [52] as well as nicotine, volatile organic compounds, and fine and ultrafine particles [53,54]. Metal and silicate particles, some of which are at higher levels than in conventional cigarettes, have been detected in e-cigarette aerosol resulting from degradation from the metal coil used to heat the solution [55]. There are limited data on the human health effects of e-cigarette emissions. According to the 2018 National Academy of Sciences comprehensive report on e-cigarettes, there is “conclusive evidence” that most e-cigarette products emit “numerous potentially toxic substances.” The report also found “substantial evidence” that e-cigarette aerosol “can induce acute endothelial cell dysfunction” and can “promote formation of reactive oxygen species/oxidative stress” [4]. Whether these findings will correlate with significant long-term clinical outcomes is to be determined, but the potential implications are concerning.

E-cigarette refill liquid presents a poisoning risk, particularly for small children, who may see the colorful bottles as toys or candy. Symptoms of nicotine poisoning include nausea, vomiting, dizziness, lethargy, dehydration, hypertension, arrhythmia, and even death. Even 5 mL of a 1.8% nicotine solution may be lethal to a toddler [56].

## 4. Public Policy Approaches to Traditional Tobacco and Implications for E-Cigarettes

Population-based policy interventions have proven to be critical in reducing cigarette smoking and smoking-related disease and death among US children and adults. Such legislative and regulatory interventions include: limiting youth access to tobacco products, restricting tobacco advertising, reducing secondhand smoke (SHS) exposure through clean air laws (including in workplaces, bars, restaurants, schools, child care facilities, parks, entertainment venues, and other public facilities), increasing taxes on tobacco products, changing the popular image of tobacco use (such as with the release of the first Surgeon General’s report, the Truth campaign, [57,58,59], and mass-media and antismoking campaigns) [60], and increasing access to smoking cessation services [25]. Few of these evidence-based approaches have been applied to public policy around e-cigarettes, and, in many ways, the current policy landscape regarding e-cigarettes resembles that of traditional combustible tobacco from 30 years ago. Even if e-cigarettes are recommended by some as a means to control tobacco-related health consequences for adults and established smokers, they still represent a significant public health burden in need of further regulation, particularly if they cause more adolescents and adults to begin harmful combustible tobacco use or prevent fewer people from quitting tobacco use [17]. The same interventions that have worked for youth smoking reduction should be applied to e-cigarettes. Below, we review tobacco control policy efforts, including their potential and challenges regarding managing and mitigating the emergence of e-cigarettes.

### 4.1. Tobacco 21 Applied to All Tobacco Products

A national campaign in the U.S. to raise the purchase age of tobacco from 18 to 21 has gained momentum over the last decade—an effort called Tobacco 21. More than 90% of adult cigarette smokers begin when they are under the age of 18 and more than 60 percent of high school users become adult smokers [25]. Communities that raise the minimum tobacco sales age to 21 see a greater decline in youth smoking than communities without such ordinances. A study in Needham, Massachusetts (the first town in the U.S. to raise the minimum tobacco sales age to 21 in 2005), found that the percentage decline in high school student smoking rates in Needham was more than double that of the surrounding communities in which the purchasing age was still 18 (6% vs. 3% decrease) [61]. Furthermore, a 2015 National Academies of Sciences report found that increasing the minimum tobacco sales age to 21 across the country could lead to a 12% reduction in smoking rates, with 249,000 fewer premature deaths and 45,000 fewer deaths from lung cancer [62].

As of February 2019, more than 425 cities and counties in 23 states across the U.S., including statewide policies in Hawaii, California, Maine, Massachusetts, Oregon and New Jersey, have raised the minimum tobacco sales age to 21 [63]. These efforts have broad support and are gaining momentum [64]. Three out of four adults favor raising the minimum tobacco sales age to 21, including almost seven out of ten current smokers [65]. Given the evidence and public support for Tobacco 21 as a means of preventing adolescent initiation of tobacco use, it would make sense—indeed, seem to be imperative—to include e-cigarettes in Tobacco 21 legislation.

### 4.2. Banning Flavors in Tobacco Products

E-cigarette solutions are often flavored, with thousands of unique flavors advertised [66,67]. Availability of flavors is among the most prominently cited reasons for youth e-cigarette use [68,69,70]. Popular options, including fruit, candy, and dessert, are particularly appealing to children and youth [66,67], more appealing than tobacco-flavors [34,71]. Further, adolescents perceive that e-cigarettes with flavors are less harmful than those with tobacco flavors, [34] creating a potential misperception that e-cigarettes with flavors do not contain nicotine [35].

Based on evidence that cigarettes flavored with candy and fruit encourage youth experimentation, regular use and addiction, [72,73,74] the Family Smoking Prevention and Tobacco Control Act of 2009 banned all cigarettes containing flavors—with the exception of tobacco and menthol [75]. The ban appears to be working, as it has been associated with a decrease in the number of cigarettes smoked among youth by 58% and the likelihood of smoking cigarettes overall in this age group by 17% [74]. These data are encouraging and suggest that this approach should also be applied to e-cigarettes. As of this publication date, however, with no restrictions on flavored e-cigarettes in general in the U.S., child-friendly flavors are still available and marketed to youth.

### 4.3. Advertising Bans

Tobacco promotion and advertising is an important cause of tobacco use initiation and escalation among youth [72]. Television and radio advertising of tobacco have been prohibited in the U.S. since 1971, and the Family Smoking Prevention and Tobacco Control Act of 2009 established, through the FDA Center for Tobacco Products, new advertising and promotions restrictions [75]. These new regulations, however, do not extend the congressional ban on cigarette TV advertisements to e-cigarettes or limit promotion tactics and the use of flavored liquids, all of which increase the appeal of these products to youth. E-cigarette companies market their products to children and adolescents by promoting flavors and using a wide variety of media channels—approaches used (with success) by the tobacco industry to market conventional tobacco products to youth [51]. E-cigarette companies, almost all of which are either owned or have an investment stake by major tobacco companies, use promotional tactics including: television advertisements targeted to stations with clear youth appeal [76]; advertisements at the point of sale at retail stores [77]; product web sites and social media [78]; targeted advertisements through search engines and web sites that focus on music, entertainment, and sports [79]; and sponsorships and free samples at youth-oriented events [51]. The use of social media and youth influencers, in particular, seems to be the primary mechanism through which e-cigarette promotion has occurred. For example, while Juul appears to have spent little to no money on traditional advertising, it has had one of the most effective social media campaigns which help promote its product across the nation [80,81,82]. Many of these e-cigarette methods of advertising are illegal for conventional cigarettes, precisely because such tactics promote youth initiation and ongoing tobacco product use [72].

Meanwhile, e-cigarette advertising has effectively reached youth. In 2016, 78.2% of middle and high school students—20.5 million youth—were exposed to e-cigarette advertisements from at least one source [83]. Exposure to advertisements increases intention to use e-cigarettes among adolescent nonusers, [84] and is associated with current e-cigarette use, [85] even in a dose-dependent fashion, as increasing exposure is associated with increased odds of use [86,87]. The increased use of and exposure to e-cigarettes among youth, combined with dramatic increases in advertising, have serious potential to undermine successful efforts to deglamorize, restrict, and decrease the use of tobacco products [17]. Incorporating and extending tobacco advertising restrictions to include bans on e-cigarette product advertising and promotion in forms that are accessible to children and adolescents would be a crucial, evidence-based measure to decrease youth e-cigarette use.

### 4.4. Clean Indoor Air Laws

According to both the Surgeon General and World Health Organization, there is no risk-free level of exposure to secondhand tobacco smoke [25,88]. The only effective measure to prevent exposure is the total elimination of smoking in indoor environments [88]. Following those evidence-based conclusions, many cities and states in the U.S. and countries around the world enacted comprehensive smoke-free legislation banning smoking in all indoor public places. Many of those laws also include outside areas near the entrances to indoor areas. The spread of the smoke-free movement and the banning of smoking indoors is lauded as one of the biggest achievements in public health in the first decade of the 21st century, protecting hundreds of millions of people from involuntary exposure to secondhand smoke both in the U.S. and worldwide [89,90].

E-cigarettes were initially advertised as a form of tobacco that could circumvent existing smoke-free legislation, [91] with initial confusion as to whether existing smoke-free legislations also apply to e-cigarettes [92]. Increasingly, smoke-free legislations banning combustible tobacco cigarette smoking in indoor public places have been amended to expand their coverage to e-cigarettes [91]. Many exceptions exist. For instance, vaping is allowed in e-cigarette shops and also in venues that hold vaping conventions (even if the use of e-cigarettes is banned in those venues during other events) [93]. The U.S. Surgeon General called on states and localities to include e-cigarettes in smoke-free policies [5]. In order to protect the public from both secondhand smoke and secondhand aerosol, the Surgeon General emphasized that smoke-free policies should be modernized to incorporate e-cigarettes, an approach that “will maintain current standards for clean indoor air, reduce the potential for renormalization of tobacco product use, and prevent involuntary exposure to nicotine and other aerosolized emissions from e-cigarettes” [51]. As of 1 October 2018, 789 municipalities, 12 states, and two territories include e-cigarettes as products that are prohibited from use in 100% smoke-free environments [94].

### 4.5. Addiction Treatment, Community Involvement and Advocacy

At the time of this writing, there are no evidence-based standards for treating nicotine addiction in children and adolescents. Nicotine replacement therapy (gum, patch), a mainstay of treatment for adult smokers, has traditionally been used only on a case-by case basis with adolescents, as evidence is less robust for efficacy in this age group [95]. However, with over 20% of high school students currently using nicotine-containing products, there is an imperative to determine recommendations and treatments. Currently, the FDA has called for input from a broad audience, including pediatricians and other health-care providers, and the research and legislative community, to address this issue. Additionally, as e-cigarette use has come into the national spotlight, parents, teachers and other community members have looked for guidance and input in addressing these issues locally. Educational sessions, open discussions and “zero-tolerance” policies, are just a few of the different approaches we have seen in our communities [96].

## 5. E-Cigarettes and Efforts of Transnational Tobacco Companies

The rapid increase in e-cigarette popularity extends across the globe: as of 2014, more than half of the world’s population lives in countries in which e-cigarettes are available [97]. Regulations and attitudes towards e-cigarettes vary considerably, depending upon whether e-cigarettes are considered a tobacco product (in need of regulation) or a reduced-harm product (to be promoted over combusted tobacco products). Some countries, such as Brazil and Uruguay, ban the products completely. Others ban e-liquids, or restrict advertising or tax e-cigarettes. In some countries, including China (where 95% of e-cigarettes are made), there are minimal restrictions in place [98]. At the 2014 meeting of the World Health Organization’s Framework Convention on Tobacco Control (FCTC), no consensus was reached regarding a unified international approach to regulation. However, the group did delineate four regulatory objectives: (1) Prevention of e-cigarettes by youth, pregnant women and non-smokers; (2) Minimizing the potential health risks of e-cigarettes; (3) Prohibition of false advertising and promotion; and, (4) Ensuring that tobacco control efforts are not impaired by tobacco companies with interest in the e-cigarette market [98]. This last point is of particular importance, given the role the tobacco industry has historically played in promoting smoking, particularly in developing nations [99]. E-cigarettes may now be providing the tobacco industry with new opportunities to increase their customer base, particular as e-cigarette users may progress to smoking. Indeed, tobacco companies have maintained and grown a significant share of the e-cigarette market.

By promoting e-cigarettes as the healthier option, tobacco companies are working to rebrand themselves. Additionally, the industry has launched another response to the documented harms of cigarette smoking: heat-not-burn (HNB) tobacco products. Philip Morris International created and is heavily marketing its version of these products, called IQOS (I-Quit-Ordinary-Smoking), involving disposable tobacco sticks, soaked in propylene glycol, which are inserted in a holder in the HNB cigarette. IQOS is not yet sold in the United States, but in December 2016, Philip Morris submitted a modified risk tobacco product application to the US Food and Drug Administration (FDA). Advertisements claim this product releases no smoke because the tobacco leaves are heated rather than burned, with no tobacco combustion, but independent studies identify cancer-causing chemicals in the smoke emitted by HNB tobacco products [100]. One study compared the contents of IQOS with the contents of conventional cigarettes, identifying that the smoke released by IQOS contains the same harmful constituents of cigarette smoke, including volatile organic compounds near the levels found in cigarette smoke, polycyclic aromatic hydrocarbons at wide comparative ranges, and carbon monoxide [101]. Based on extensive research of cigarette smoke, all of these elements cause serious harm to human health [25]. A second study found that the aerosol from HNB tobacco products contains nicotine and similar cancer-causing chemicals as traditional cigarettes [102]. Additional reports, reviewing Philip Morris’s own data made available through the FDA application process, found IQOS is associated with significant pulmonary and immunomodulatory toxicities with no detectable differences between conventional cigarette smokers and IQOS, [103] and IQOS use, compared to cigarettes, reduces exposure to some toxicants but elevates exposure to other substances [104].

Regardless of the health concerns raised and identified by public health advocates, the tobacco industry is making considerable investment in e-cigarettes and HNB products globally. A recent analysis of this trend raises important questions regarding the companies’ intent, which “appears to be to sustain, rather than replace, cigarette sales, and to increase their influence and credibility with respect to e-cigarette policy and regulation” [105].

## 6. Conclusions

The increasing popularity and prevalence of e-cigarette use among youth threatens to reverse decades of progress in tobacco control. Concerns regarding nicotine addiction and progression to combusted tobacco smoking have fueled debate and some, but not enough, regulatory action. While long-term data are lacking, there is mounting concern regarding the toxic potential of these products, both to the user, and to those exposed to the aerosol. Implementing established evidence-based approaches to decrease tobacco use in youth should serve as a template for regulation. Failure to act now could have devastating long-term consequences.

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
