# Peer review of "Electronic Cigarettes and Youth in the United States: A Call to Action (at the Local, National and Global Levels)"

_children, 2019, doi:10.3390/children6020030_

Round 1

Reviewer 1 Report

Electronic Cigarettes and Youth: A Call to Action (At the Local, National and Global Levels)

This is an interesting, well-written manuscript which will be an important addition to the literature. Please see my comments below.

Paragraph of Brief History of e-cigarettes:

Please highlight the role of JUUL or USB-like e-cigarettes, the fastest growing product which has captured over 75% of the market share of the e-cig category since its introduction in 2014-2015. This product transformed the category and use patterns given its efficient delivery of nicotine, flavors and covert design. 

Please add in the following citations for e-cig and JUUL data:

Vallone D, Bennett M, Cuccia A, Hair E. (2018). Prevalence and correlates of JUUL use among a national sample of youth and young adults. Tobacco Control.

Cantrell J, Huang J, Greenberg M, Willett J, Hair E, Vallone D. (2018). History and Current Trends in the Electronic Nicotine Delivery Systems Retail Marketplace in the United States: 2010-2016. Nicotine Tob Res.

Koval R, Willett J, Briggs J. (2018). High-Nicotine E-Cigarettes: Benefits and Risks. Journal of the American Medical Association.

Line 132: Citations listed are from decades ago. Please update citations related to the Truth campaign:

1. Vallone D, Cantrell J, Bennett M, Smith A, Rath J, Xiao H, Greenburg M, Hair EC. (2017). Evidence of the Impact of the truth FinishIt Campaign. Nicotine and Tobacco Research.

2. Vallone DM, Greenberg M, Xiao H, Bennett M, Cantrell MJ, Rath J, Hair EC. (2017). The Effect of Branding to Promote Healthy Behavior: Reducing Tobacco Use among Youth and Young Adults. International Journal of Environmental Research and Public Health.

3. Hair E, Pitzer L, Bennett M, Halenar M, Rath J, Cantrell J, Dorrler N, Asche E, Vallone D. (2017). Harnessing Youth and Young Adult Culture Improving the Reach and Engagement of the truth® Campaign. Journal of Health Communication.

Line 133: citations of other anti-smoking campaigns

Farrelly MC, Duke JC, Nonnemaker J, et al. Association Between The Real Cost Media Campaign and Smoking Initiation Among Youths — United States, 2014–2016. MMWR Morb Mortal Wkly Rep 2017;66:47–50. DOI: http://dx.doi.org/10.15585/mmwr.mm6602a2.

Lines 191-200

Please the role of social media and youth influencers as a primary mechanism through which e-cigarette promotion has occurred. JUUL has spent little to none on traditional advertising, but had one of the most effective social media campaigns which help promote its product across the nation. See papers below.

https://www.cnn.com/2018/12/17/health/juul-social-media-influencers/index.html

https://www.thefix.com/did-juul-use-young-people-create-e-cig-buzz-social-media

https://news.gsu.edu/2018/06/04/study-shows-savvy-social-media-campaigns-spur-growth-in-juul-device-use/

Author Response

Point 1: Paragraph of Brief History of e-cigarettes: Please highlight the role of JUUL or USB-like e-cigarettes, the fastest growing product which has captured over 75% of the market share of the e-cig category since its introduction in 2014-2015. This product transformed the category and use patterns given its efficient delivery of nicotine, flavors and covert design.

Response 1: Addressed, see lines 64-67

Point 2: Please add in the following citations for e-cig and JUUL data:

Vallone D, Bennett M, Cuccia A, Hair E. (2018). Prevalence and correlates of JUUL use among a national sample of youth and young adults. Tobacco Control.

Cantrell J, Huang J, Greenberg M, Willett J, Hair E, Vallone D. (2018). History and Current Trends in the Electronic Nicotine Delivery Systems Retail Marketplace in the United States: 2010-2016. Nicotine Tob Res.

Koval R, Willett J, Briggs J. (2018). High-Nicotine E-Cigarettes: Benefits and Risks. Journal of the American Medical Association.

Response 2: Addressed and added, see lines 67 and references 07/02/2019 08:08:00

Point 3: Line 132: Citations listed are from decades ago. Please update citations related to the Truth campaign:

1. Vallone D, Cantrell J, Bennett M, Smith A, Rath J, Xiao H, Greenburg M, Hair EC. (2017). Evidence of the Impact of the truth FinishIt Campaign. Nicotine and Tobacco Research.

2. Vallone DM, Greenberg M, Xiao H, Bennett M, Cantrell MJ, Rath J, Hair EC. (2017). The Effect of Branding to Promote Healthy Behavior: Reducing Tobacco Use among Youth and Young Adults. International Journal of Environmental Research and Public Health.

3. Hair E, Pitzer L, Bennett M, Halenar M, Rath J, Cantrell J, Dorrler N, Asche E, Vallone D. (2017). Harnessing Youth and Young Adult Culture Improving the Reach and Engagement of the truth® Campaign. Journal of Health Communication.

Response 3: Addressed and updated, see lines 158 and references

Point 4: Line 133: citations of other anti-smoking campaigns

Farrelly MC, Duke JC, Nonnemaker J, et al. Association Between The Real Cost Media Campaign and Smoking Initiation Among Youths — United States, 2014–2016. MMWR Morb Mortal Wkly Rep 2017;66:47–50. DOI: http://dx.doi.org/10.15585/mmwr.mm6602a2.

Response 4: Addressed and updated, see line 159 and references

Point 5: Lines 191-200

Please the role of social media and youth influencers as a primary mechanism through which e-cigarette promotion has occurred. JUUL has spent little to none on traditional advertising, but had one of the most effective social media campaigns which help promote its product across the nation. See papers below.

https://www.cnn.com/2018/12/17/health/juul-social-media-influencers/index.html

https://www.thefix.com/did-juul-use-young-people-create-e-cig-buzz-social-media

https://news.gsu.edu/2018/06/04/study-shows-savvy-social-media-campaigns-spur-growth-in-juul-device-use/

Response 5: Addressed. Added original reports for the 2nd and 3rd suggested citation, see lines 225-229.

Reviewer 2 Report

General comments

This is a review which deals with health effects of the electronic cigarettes in youth and the possible implications of extending the tobacco control policies to the electronic cigarettes.

The paper is well-written and structured. 

Specific comments

Major

The paper deals with epidemiological data and tobacco control policies exclusively from the United States. This should be indicated in the title.

The need to extend the current tobacco control policies to the electronic cigarettes in order to protect the youth health is clearly underlined in different parts of the paper (see for example: page 3, lines 136-137, "The same interventions that have worked for smoking reduction should be applied to e-cigarettes."; page 4, lines 157-158, "it would make sense - indeed, seem to be imperative - to include e-cigarettes in Tobacco 21 legislation". However, in some countries (e.g. United Kingdom) e-cigarettes are recommended as a way to control tobacco-related health consequences at public health level. This critical point, including the possible implications for youth, should be discussed.

Updated studies on youth perception of health-risks and addictiveness of electronic cigarettes with nicotine should be reviewed.

Page 2, lines 44-61. This two paragraphs do not deal with the e-cigarettes history but do introduce background and aim of the review, which instead should be presented as a dedicated subsection. The aim of the review should be clearly stated.

Minor

Page 2, lines 50-51. It should be specified: "in the United States".

Author Response

Point 1: The paper deals with epidemiological data and tobacco control policies exclusively from the United States. This should be indicated in the title.

Response 1: Addressed, see title

Point 2: - The need to extend the current tobacco control policies to the electronic cigarettes in order to protect the youth health is clearly underlined in different parts of the paper (see for example: page 3, lines 136-137, "The same interventions that have worked for smoking reduction should be applied to e-cigarettes."; page 4, lines 157-158, "it would make sense - indeed, seem to be imperative - to include e-cigarettes in Tobacco 21 legislation". However, in some countries (e.g. United Kingdom) e-cigarettes are recommended as a way to control tobacco-related health consequences at public health level. This critical point, including the possible implications for youth, should be discussed.

Response 2: Agreed and addressed, see lines 162-166

Point 3: Updated studies on youth perception of health-risks and addictiveness of electronic cigarettes with nicotine should be reviewed.

Response 3: Addressed, see lines 108-110

Point 4: Page 2, lines 44-61. This two paragraphs do not deal with the e-cigarettes history but do introduce background and aim of the review, which instead should be presented as a dedicated subsection. The aim of the review should be clearly stated.

Response 4: Addressed, see lines 46 and 70-73

Point 5: Page 2, lines 50-51. It should be specified: "in the United States".

Response 5: Addressed, see lines 52-53

Round 2

Reviewer 2 Report

The authors have adequately addressed the comments from reviewers, the paper has improved and, in the opinion of this reviewer, it is now suitable for publication.